# Quantitative Proteomics and Network Analysis of Differentially Expressed Proteins in Proteomes of Icefish Muscle Mitochondria Compared with Closely Related Red-Blooded Species

**DOI:** 10.3390/biology11081118

**Published:** 2022-07-26

**Authors:** Gunjan Katyal, Brad Ebanks, Adam Dowle, Freya Shephard, Chiara Papetti, Magnus Lucassen, Lisa Chakrabarti

**Affiliations:** 1School of Veterinary Medicine and Science, University of Nottingham, Sutton Bonington LE12 5RD, UK; gunjan.katyal@nottingham.ac.uk (G.K.); brad.ebanks@nottingham.ac.uk (B.E.); freya.shephard@nottingham.ac.uk (F.S.); 2Department of Biology, Bioscience Technology Facility, University of York, York YO10 5DD, UK; adam.dowle@york.ac.uk; 3Biology Department, University of Padova, Via U. Bassi, 58/b, 35121 Padova, Italy; chiara.papetti@unipd.it; 4Alfred Wegener Institute, 27570 Bremerhaven, Germany; magnus.lucassen@awi.de; 5MRC-Versus Arthritis Centre for Musculoskeletal Ageing Research, Liverpool L7 8TX, UK

**Keywords:** icefish, proteomics, mitochondria, muscle, network analysis, notothenioid

## Abstract

**Simple Summary:**

Antarctic icefish are unusual in that they are the only vertebrates that survive without the protein haemoglobin. One way to try and understand the biological processes that support this anomaly is to record how proteins are regulated in these animals and to compare what we find to closely related Antarctic fish that do still retain haemoglobin. The part of the cell that most clearly utilises oxygen, which is normally transported by haemoglobin, is the mitochondrion. Therefore, we chose to catalogue all the proteins and their relative quantities in the mitochondria (pl.) from two different muscle types in two species of icefish and two species of red-blooded notothenioids. We used an approach called mass spectrometry to reveal relative amounts of the proteins from the muscles of each fish. We present analysis that shows how the connections and relative quantities of proteins differ between these species.

**Abstract:**

Antarctic icefish are extraordinary in their ability to thrive without haemoglobin. We wanted to understand how the mitochondrial proteome has adapted to the loss of this protein. Metabolic pathways that utilise oxygen are most likely to be rearranged in these species. Here, we have defined the mitochondrial proteomes of both the red and white muscle of two different icefish species (*Champsocephalus gunnari* and *Chionodraco rastrospinosus)* and compared these with two related red-blooded Notothenioids (*Notothenia rossii*, *Trematomus bernacchii*). Liquid Chromatography-Mass spectrometry (LC-MS/MS) was used to generate and examine the proteomic profiles of the two groups. We recorded a total of 91 differentially expressed proteins in the icefish red muscle mitochondria and 89 in the white muscle mitochondria when compared with the red-blooded related species. The icefish have a relatively higher abundance of proteins involved with Complex V of oxidative phosphorylation, RNA metabolism, and homeostasis, and fewer proteins for striated muscle contraction, haem, iron, creatine, and carbohydrate metabolism. Enrichment analyses showed that many important pathways were different in both red muscle and white muscle, including the citric acid cycle, ribosome machinery and fatty acid degradation. Life in the Antarctic waters poses extra challenges to the organisms that reside within them. Icefish have successfully inhabited this environment and we surmise that species without haemoglobin uniquely maintain their physiology. Our study highlights the mitochondrial protein pathway differences between similar fish species according to their specific tissue oxygenation idiosyncrasies.

## 1. Introduction

Mitochondria are crucial organelles that produce ATP via oxidative phosphorylation, a process that involves the transfer of electrons between multi-subunit complexes (I to IV) of the electron transport chain (ETC), resulting in the reduction of molecular oxygen. These organelles are also known for their involvement in many vital cellular activities such as apoptosis, calcium homeostasis, and regulation of cell homeostasis. Due to this, there has been an increasing interest in the structure and function of mitochondrial proteins [1,2,3,4]. Efforts have been made to decipher the structure, assembly process, coupling mechanism, and associated pathologies of respiratory chain complexes [5,6].

Haemoglobin (Hb) synthesis requires a coordinated production of both haem and globin. Hb is a multi-subunit globular molecule made up of four polypeptide subunits, two alpha and two beta subunits. Each of the four subunits has a haem moiety that contains iron [7]. The prosthetic group haem is synthesised in a series of steps shuttling between the mitochondrion and the cytosol of immature erythrocytes [8]. To understand the role of haemoglobin, it can be useful to examine eukaryotic systems that express this important protein at different levels. There is a wide range of evidence that suggests Hb has dynamic locations in cell, neurons, endothelial cells, mitochondria, and vascular expression [9,10,11,12,13]. Previously, we have shown that haemoglobin proteins are located in the mitochondrion and both direct and indirect links have been established between mitochondria function and Hb expression [10,14,15].

Interestingly, a group of vertebrates known as Antarctic icefish are “null-mutants” for haemoglobin that have some likely relevant cellular modifications, including a high mitochondrial density [16]. Antarctic icefish/white-blooded fish, (subfamily Channichthyidae, family Nototheniidae, suborder Notothenioidei [17]) are the only known vertebrates that do not possess functional haemoglobin genes and red blood cells (RBCs), in stark contrast to all the others that depend upon Hb to get oxygen to tissues and cells, via RBCs [18,19,20]. The loss of Hb in the Antarctic icefish is postulated to be a mutational process, resulting in the loss of the β-globin (hbb) gene and partial omission of the α-globin (hba) gene from α-/β-globin, causing the locus to become functionally inactive. Fifteen out of the sixteen icefish species are known to retain only a 3′ fragment of an α-globin gene [21]. The sixteenth species, *Neopagetopsis ionah*, retains an intact but unexpressed hba gene fused to two β-globin pseudogenes [22].

As might be expected, some proteins have been shown to be altered in icefish and reasons have been suggested for these differences. Previously, studies have shown changes in the iron transporting proteins such as transferrin, ceruloplasmin, and ferritin [23]. Not all notothenioid species express myoglobin (Mb), an intracellular oxygen-binding protein in the muscle. In total, six of the sixteen icefish species do not express Mb in their heart ventricles, a loss that occurred through four different mutational events [24]. Icefish have large hearts [25], which do not contain the mitochondrial creatine kinase [26]. They also have high mitochondrial densities, which is postulated to counter the effects of cold temperature [27]. The high density of mitochondria rich in lipids serves as a pathway that enhances oxygen storage and diffusion, which compensates for the lack of Hb and Mb [28]. An observed increase in mitochondrial phospholipids may be due to an upregulation in the glycerol-lipid synthesis pathway. There is still a lot to be learned about protein networks and pathways in icefish [29].

Proteomics can be used to identify protein-protein interactions, which in turn influence protein expression or regulation [30]. In silico methods and web servers have been developed to predict the function and structure of proteins [31]. Although large-scale mitochondrial comparative proteomic data have been accumulated, mitochondrial proteomics still faces the challenge of how to investigate the functions of the identified mitochondrial proteins and how to build mitochondria specific signalling networks. An integrative network analysis approach can accommodate information from PPIs and proteomics and bridge the gap between the two. Hence, using an interactive network approach could lay the foundation for a better understanding of mitochondrial changes in icefish [32].

Here, we establish for the first time the proteomes of red muscle mitochondria (RMM) and white muscle mitochondria (WMM) from four species of the suborder Notothenioidei: two icefish, *Champsocephalus gunnari* (*C. gunnari*) devoid of Hb and Mb (the loss of Hb and Mb completely) and *Chionodraco rastrospinosus* (*C. rastrospinosus*), devoid of Hb but with tissue specific expression of Mb that is only expressed in the hearts of these species, are compared to their closely related red-blooded species *Notothenia rossii* (*N. rossii*) and *Trematomus bernacchii* (*T. bernacchii*), both the species belonging to the Nototheniidae, expressing Hb, and having tissue specific expression of Mb only in heart ventricles. Mammalian skeleton muscles are mainly composed of two kinds of fibres, white-fast twitch type that makes the white muscle and red-slow twitch type that makes the red muscle [33]. White muscle requires a greater capacity for anaerobic energy production to meet the demands of the fast-twitch fibres [34]. In contrast to mammalian white muscle tissue, the central regions of icefish red muscle contain numerous mitochondria [16]. In establishing these proteomes, we can understand how the mitochondrial proteome has adapted to the loss of this protein and potentially understand the role and interaction pathways of haemoglobin in the context of mitochondrial biology.

## 2. Methodology

**Antarctic Fish Muscle tissue:** SVMS Clinical Ethical Review, University of Nottingham, (ref # 2744 190509). White and red muscle samples of Antarctic notothenioid fish, *N. rossii*, *T. bernacchii* (red-blooded species—Cruise PS112, Weddell Sea in 2013–2014), *C. gunnari* and *C. rastrospinosus* [icefish species—Cruise ANTXXVIII (PS79), Antarctic Peninsula in 2012]. *N. rossii* and *T. bernacchii* have 30–45% of haematocrit and 18–28% haematocrit respectively.

Ethics Statement: The proposal for the Antarctic Fish project was approved by Veterinary School′s Clinical Ethical Review panel, University of Nottingham, with responses from at least four members of the panel. No ethical issues were raised by the panel. The Ethical project Number issued: 2744 190509.

**Subcellular fractionation:** The white and red muscle tissues were stored at −80 °C and while obtaining the fractions were put on ice throughout. The tissues were cut with a sterile blade and then homogenised using Dounce homogeniser, in 2 mL mitochondria extraction buffer (50 mM Tris-Cl pH 7.4, 100 mM KCl, 1.5 mM MgCl2, 1 mM EGTA, 50 mM HEPES and 100 mM sucrose). The homogenized mixture was centrifuged at 800 rpm for 10 min at 4 °C to remove the insoluble fraction, and the first supernatant was centrifuged at 1000 rpm to pellet nuclear fraction. The supernatant obtained from the second centrifuge, was centrifuged at 10,500 rpm at 4 °C for 30 min to obtain the mitochondrial fraction. The supernatant which contained the cytosolic fraction was transferred into a separate tube and the pellet contained the mitochondrial fraction. The quality of the crude fractions was confirmed using standard western blotting techniques with nuclear, mitochondrial and cytoplasmic markers (Histone H3, ab1791 (Abcam) Rb pAB; COX IV ab16056 (Abcam) Rb Ab; and GAPDH, Sigma G9545 Rb respectively) as described previously specific blots or this study can be found [15]. A majority of the proteins identified are associated with mitochondrial processes, confirming the successful fractionation.

**Liquid Chromatography-Mass spectroscopy proteomic analysis** (**LC-MS/MS**)**:** Complete mass spectrometry data sets and proteomic identifications are available to download from MassIVE (MSV000089644), [doi:10.25345/C5PZ51Q9F] and ProteomeXchange (PXD034498).

Red Muscle: The extracted mitochondrial fractions of three specimens per species (*C. gunnari:* samples 639, 673, 690; *C. rastrospinosus*: sample 1005, 1020, 1355; *N. rossii*: 4–10, 4–27, 5–14; *T. bernacchii*: samples 7–23, 7–56, 8–24) were captured in gel and sent to the Metabolomics and Proteomics Lab (University of York, York, UK) for Liquid Chromatography-Mass spectroscopy (D-100) proteomic analysis. A label-free, intensity-based quantification for comparing relative protein amounts between samples approach was used. Mass spectrometry data were analysed using PEAKSX software. The mapped ion areas were used as metric for significance testing for changes in the abundances between groups by using the PEAKSQ interpretation of the significance of the B model. These were converted into relative percent of the total ion area for analysis. The PEAKSQ significance values have been multiply-test-corrected using the Hochberg and Benjamin FDR approach (*q* < 0.01).

Protein identification: The spectra from PEAKSX studio were searched against the combined NCBI deposited proteins from *Notothenia coriiceps* (32,361 sequences; 15,554,893 residues)*, Chaenocephalus aceratus* (223 sequences; 59,314 residues)*, Dissostichus mawsoni* (210 sequences; 61,335 residues) and *Eleginops maclovinus* (193 sequences; 53,595 residues), in addition to 115 common proteomic contaminant proteins. Protein identifications were filtered to achieve <1% false discovery rate (FDR) as assessed against a reverse database. Identifications were further filtered to require a minimum of two unique peptide identifications per protein group.

White Muscle: The extracted mitochondrial fractions of three specimens per species (*C. gunnari:* 675, 708, 641 samples; *C. rastrospinosus*: 1006, 1021, 1386 samples; *T. bernacchii*: 7–37, 7–54, 8–22 samples; *N. rossii*: 4–25, 4–42, 4–59 samples) were captured in gel and sent to the Metabolomics and Proteomics Lab (University of York, UK) for Liquid Chromatography-Mass spectroscopy (D-270) proteomic analysis. The same approach was applied for analysing the mass spectrometry data for the white muscle mitochondria samples and was matched against the previously mentioned NCBI deposited proteins.

From the proteomic data, we used ‘number of spectral matches’ as it shows the best metric for the presence and absence of a predicted protein. For the quantitative analysis of the changes in protein proportions, we used the relative percent of total ion map area for comparison of the proteins.

Proteins that were individually low in abundance or higher in abundance for each species were characterised using the number of spectral matches and Ion Map Area (Supplementary Appendix A). The proteins from those lists present in species-specific quantities were grouped and used for comparing the abundance of proteins in red-blooded species (*N. rossii* & *T. bernacchii*) samples and with icefish species (*C. gunnari* & *C. rastrospinosus*) samples. This also took in account the proteins that followed a similar trend in their amounts, i.e., more in abundance in icefish in comparison to red-blooded species and vice versa.

There is no proteomic or genomic data available for these samples so searching against other species was the only option. To help ameliorate issues such as the case of divergent sequences of proteins or/and quantification data of identified peptides, three rounds of searches were performed—initially against Notothenia coriiceps, Chaenocephalus aceratus, Dissostichus mawsoni, and Eleginops maclovinus, which were publicly available using conventional parameters; subsequently, searching was expanded to include 313 common PTMs and single point amino acid substitutions. The inclusion of single point substitution can allow for better coverage between non-identical species. The similarity in mitochondrial proteins between these species and this coupled strategy helps elevate that issue. The similarity in mitochondrial proteins between these species, coupled with the search strategy applied, means that this divergence in species does not appear to be too dramatic in these samples

**STRING Network analysis and Clustering**: The list of proteins that were found to follow a definite pattern in the proteome of the icefish in comparison to the red-blooded fish were checked as separate groups in the STRING database (http://string-db.org; accessed on 20 February 2021). The protein IDs of the selected protein was extracted from the proteomics data and the respective FASTA sequences were extracted. The multiple FASTA sequence of the proteins was searched against the closest available species on the STRING db, *Danio rerio* (zebrafish) using the multiple sequence. The STRING networks were generated with a medium confidence (0.400) and 5% FDR. Network analyses were visualised in Cytoscape_v3.8.0 [35] and to obtain the top hub proteins, a molecular complex detection plug-in (MCODE) was used to obtain the modules. The criteria used were Degree cut-off = 2, node score cut-off = 0.2, k-core = 2 and max Depth = 100 [36]. The top modules (score > 4) were selected for graphical representation. The hub nodes (proteins) in the PPI were also analysed by their topological relevance using the Cytoscape plugin CentiScaPe [37] (Appendix A) with default options. The plots were generated using the plot by node option and supports the importance of those highlighted hubs, specifically pointing out the shortest path betweenness and centroid values.

Cytoscape with the GeneMANIA plugin was used to identify the genes most related to the groups of gene sets to form a network of functional genes based on their interaction, such as co-expression, physical-interaction, and shared protein domains. GeneMANIA, a plug-in for Cytoscape, predicts the function of the identified gene sets by using a ‘guilt-by-association’ approach informed by functional networks from multiple organisms [38]. The gene symbols for the proteins were inputted with default parameters using in-built *D. rerio* gene information, which have been collected from GEO, BioGRID and organism-specific functional genomic data sets. The pathway enrichment was analysed for GO terms ‘biological significance’ and ‘associated metabolic pathways in KEGG’. The enrichment database fishENRICHR was run using gene symbols (Table 1, Table 2, Table 3 and Table 4); these were sorted according to the *p* values (*p* < 0.05, probability of any gene belonging to any set) [39,40].

Graphical representation: R package ggplot2 v.3.5.1. was used for generating heatmaps and volcano plots.

## 3. Results

### 3.1. LC/LC-MS Data for RMM and WMM

The purpose of the study was to identify mitochondrial and mitochondrial-associated proteins in icefish and to understand how the loss of haemoglobin affects the proteome. LC-MS/MS data were analysed using PEAKSX for reliable matching to the available sequence database and the data were filtered to a 1% false discovery rate by at least two unique peptides for each protein group (Figure 1(A1,B1)). For red muscle mitochondria (RMM), 1148 proteins were identified belonging to unique protein groups; this contrasts with white muscle mitochondria (WMM), where 429 proteins were identified. Haemoglobin α, β and γ were only identified in the red-blooded fish confirming sample specificity. The two types of muscle are faced with very different energetic demands; however, previously it has been shown that there are no large differences in mitochondrial protein expression when surveying different porcine muscle tissues [41].

Group 1 consisted of the proteins that were significantly more abundant, and group 2 consisted of less abundant proteins in the two icefish species when compared with the red-blooded species (Table 1, Table 2, Table 3 and Table 4). The reported significance value is the –log10 *p* value, with the null hypothesis being that the protein is of equal abundance in all samples. The higher the significance value the greater the probability that the protein is not equal in abundance in all groups. The volcano plot distinctively shows the significant differentially expressed proteins (DEPs) (*q* < 0.05) that were downregulated (green) and upregulated (red). The WMM had fewer proteins that were not significant when compared to the RMM. A heat map that includes the differentially expressed proteins (DEPs) in red muscle (Figure 1(A2)) and white muscle (Figure 1(B2)) mitochondria illustrates that protein expression profiles are characteristic for the haemoglobin-less species when compared with red-blooded groups. The mapped ion areas were converted to relative percent values by weighting each protein equally to point out differences between groups.

There was a common increase in abundance in the complex V proteins and a group of ribosomal proteins (more evidently in the white muscle) in the icefish in both the muscle mitochondria tissues. A few mitochondrial import proteins were also seen to be increased such as ADP/ATP translocases, voltage-dependent anion channels, and heat shock proteins. A common decrease in abundance in the proteins for the icefish were seen in Hb and haem/Hb-associated proteins such as cytochrome c, transferrin, haem oxygenase 2 and hemopexin. Apart from those, a decrease in the abundance was seen in muscle proteins such as creatine kinase, troponin, titin, and myosin heavy chain.

The two groups of proteins from both the tissues RMM and WMM were mapped into protein-protein interaction (PPI) networks constructed using STRINGdb. The enrichment *p*-value for each of the three PPI networks is lesser than10^−16^, indicating that proteins share more interactions than would be expected for a random set of proteins of similar size drawn from the proteome and suggesting at least partial biological connection as a group [42]. The network was retrieved and analysed using Cytoscape software, which allowed us to visualize and analyse molecular interaction networks [43] (Figures 2 and 4) [38].

### 3.2. Analysis of Proteins More in Abundance in Icefish

Forty-three proteins were differentially expressed with higher abundance in the RMM of the two icefishes compared to the red-blooded nonfamilial species. Fifty-seven proteins were significantly higher expressed in the WMM (Table 1 and Table 2). Most proteins that were seen upregulated in both RMM and WMM were found to be part of the complex V, ribosomal and proteosome machineries in addition to a common upregulated expression of malate dehydrogenase and Fragile X mental retardation isoform 2 protein. In RMM, components of complex V of the electron transport chain and proteins involved in transportation across mitochondria were observed. The upregulated proteins of WMM were involved in the citrate cycle and carbon metabolism.

GeneMANIA (based on zebrafish) analysis for interactions of the proteins with increased abundance in icefish produced one network per each muscle tissue (Figure 2A,B). In RMM, co-expression occupied 98.66% of the interactions seen in the network; shared protein domains occupied 1.13%; physical interaction, 0.21%. The network showed shared domains among proteins slc25a5, slc25a6, slc25a20, vdac3 and slc25a12, and, between atp5a1 and atp5b. The network showed physical interactions between atp5a1, atp5o, predicted cyc1 and coq9. In WMM, co-expression occupied 98.46% of the interactions seen in the network; physical interactions occupied 1.431%; and shared protein domains occupied 0.11%. The network showed physical interactions between ribosomal proteins and separately also between proteasome proteins.

Figure 3A shows the PPI network generated using the FASTA sequences (corresponding NCBI IDs) for the proteins using STRING. The 43 identified DEPs that were more abundant in RMM were analysed and connected with a PPI enrichment *p*-value < 1.0 × 10^−16^, with 42 nodes (proteins RPL27 was not identified for organism *D. rerio*), 143 edges, and an average node-degree of 6.8. Four of forty-two DEPs (RTN4IP1, PFKMA, FRX2 and HAPLN1b) did not connect to any type of network (STRING interaction score = 0.4). Thirty-eight of the remaining DEPs were connected to networks by complex relationships. RPL23, MDH1AA, ATP5B, and VDAC3 showed network hubs highly associated with other nodes in PPI. (Appendix A). A single network was formed between DEPs ITIH4, A2ML, KNG1, F2 and APOBB. The highly connected proteins are majorly involved in energy metabolism and protein metabolism. Three distinctive clusters were seen for RMM upregulated proteins. Cluster 1 had proteins involved in ETC and oxidative phosphorylation, Cluster 2, proteins involved in cell signalling, and Cluster 3, proteins involved in fatty acid biosynthesis (Appendix A).

Functional enrichment analysis (FDR < 1.9 × 10^−2^) showed proteins involved in the TCA cycle, oxidative phosphorylation, degradation of RNA, cristae formation, mitochondrial protein import, and carbon metabolism (See Figure 3A), using the STRINGdb information provided under KEGG and Reactome pathways.

Using FishENRICHR [39,40], the most used GO Terms for different biological processes were: GO:0019674, NAD metabolic process; GO:0070306 lens fibre cell differentiation; GO:0006754 ATP biosynthesis; GO:0006839 mitochondrial transport; GO:0045898 regulation of RNA polymerase II transcriptional pre-initiation complex assembly. (See Figure 3(C1,C2)).

Using the same steps as for RMM, a PPI network was generated for WMM DEPs (Figure 3B). As for the WMM, the intra network connections were strongest (PPI enrichment *p*-value < 1.0 × 10^−16^, with fifty-six nodes, 382 edges, and an average node-degree of 13.6). Nine of the fifty-six DEPs did not connect to any network (RTNA1, MYOZ1B, HSPB1, KLH41B, TUBA8L2, VTG2, ZGC:110377, PALLD, and FXR2- Table 2).

The remaining forty-seven of the differentially expressed proteins were connected to networks where differentially expressed proteins ATP5B, GNB2L1, RPL11, RPL13, RPSA, and PKMA are the major protein-hubs (See Appendix A). A single network was formed between DEPs MYLPFB, AMPD, ACTN2, and VCLA.

Three distinctive clusters were seen for WMM upregulated proteins similar to RMM. Cluster 1 had proteins of ribosome machinery, Cluster 2 proteins involved in ETC and TCA, and Cluster 3 proteins involved in fatty acid biosynthesis (Appendix A).

Functional enrichment analysis (FDR < 1.4 × 10^−2^) showed proteins involved in TCA, ribosomal proteins, downstream signalling events of B cell receptors, and L13a-mediated translational silencing of ceruloplasmin. The fishENRICHR identified, GO:0000463: “maturation of LSU-rRNA”, GO:0045727 and GO:0000470: “positive regulation of translation”, GO:0000027 and GO:0042273: “ribosomal large subunit assembly” as most common GO terms. Enriched KEGG pathways included ribosome, glycolysis, and gluconeogenesis, and the pentose pathway (pathways sorted according to *p*-values) (Figure 3(C3,C4)). The proteins involved in gluconeogenesis have previously been reported altered in their expression in rainbow trout. The study goes on to show an increase in the enzyme FB2, a key enzyme of gluconeogenesis to be increased in red muscle of the fish. On contrary we see FB2 to be increased fourfold in the white muscle tissue for icefish rather than the red muscle [44].

The protein networks between RMM and WMM differ. In RMM, there is one quite dense cluster with some weak “satellites” and another cluster with similar connectivity as seen in WMM. In WMM there are three separate clusters with good and quite similar connections.

### 3.3. Proteins with Lower Abundance in Icefish

Forty-eight proteins were found in lower quantities in RMM (ENSDARG00000030638, ca1, casq, gdh, mdh, pygm, rdh13, wu:fd55e03 were not recognised by GENEmania) and thirty-two proteins were differentially expressed in WMM in icefish compared to their red-blooded relatives (Table 3 and Table 4). As before, GeneMANIA was used to analyse the interactions and produced one network per tissue (Figure 4A,B). In RMM, co-expression accounted for 96.14% of the total interactions seen in the network; and shared protein domains occupied 3.86%. In WMM, co-expression occupied 94.76%; physical interactions occupied 5.02%; and shared protein domains occupied 0.21%. The network showed physical interactions between ribosomal proteins and nebulin and this was predicted for the neighbouring protein tropomodulin.

In RMM, PPI networks include forty-eight protein nodes (Figure 5A), nine proteins (MUT, RPN1, CKMT2A, PDPR, DHRS7CB, ALDH7A1, WU:FD55e03, USP5, PLIN3) did not connect to any type of network (STRING interaction score = 0.4). The PPI enrichment (*p*-value < 1.0 × 10^−16^) had thirty-nine nodes, fifty-three edges, and an average node degree of 2.52. The rest of the DEPs were connected to networks by complex relationships, where proteins ACTN3B, TTNA, CKMB, MYBPC1, TTNT3B, and CYCSB showed network hubs highly associated with many proteins (Appendix A). Single networks were observed between FBN2A and TFA, SLC4A1A and HBAE1. Two distinctive clusters were seen for RMM. Cluster 1 had proteins involved in striated muscle contraction, and Cluster 2 proteins involved in oxidative stress (Appendix A).

In WMM, six (ZGC:153629, HBZ, HBAA1, MYOM1A, EVA1BA and, MDH1AA) out of the thirty-two protein nodes in the network did not connect to any type of network (Figure 5B). Proteins FGA, TFA, and TTNA represent network hubs, highly associated with other proteins (Appendix A). The PPI enrichment (*p*-value < 3.49 × 10^−12^) gave thirty-two nodes with forty-five edges and an average node-degree of 2.58. Three distinctive clusters were seen for WMM downregulated proteins. Cluster 1 had proteins of actin filament regulation, Cluster 2, proteins involved in muscle contraction, and Cluster 3, proteins involved with collagen (Appendix A).

The functional enrichment analysis (FDR < 3.8 × 10^−3^) for RMM reduced proteins in icefish were involved in striated muscle contraction, erythrocytes absorbing oxygen, fibrin clot formation hemostasias, and haem-associated proteins (Figure 5A). The following GO terms for biological processes were found overrepresented: GO:0006941 and GO:0006936: “striated muscle contraction” and GO:0045214: “sarcomere organisation”. The following KEGG pathways were affected: amino acid metabolism, fatty acid degradation, and peroxisome, glyoxylate, and dicarboxylate metabolism (Figure 5(C1,C2)).

The functional enrichment analysis (FDR < 2.5 × 10^−4^) for white muscle downregulated proteins in icefish identified multiple pathways, including translational silencing proteins and signal-recognition particle SRP-dependent co-translational protein targeting, the latter being involved in binding to the endoplasmic reticulum (ER) (Figure 5B).

Based on fishENRICHR analysis, the most used GO Terms for different biological processes were: GO:0019674 NAD metabolic process, GO:0060956 endocardial cell differentiation, GO:0020027 haemoglobin metabolic process, and GO:1903512 endoplasmic reticulum to cytosol transport (Figure 5(C4)). AGE-RAGE signalling pathway metabolism, glyoxylate and dicarboxylate metabolism, amino acid metabolism, ECM-receptor interaction, and focal adhesion were identified by KEGG pathway analysis (See Figure 5(C3)).

## 4. Discussion

The paper presents the comparative analysis of the mitochondrial proteomes of white (WMM) and red muscle (RMM) of icefish that do not express haemoglobin protein and closely related red-blooded species. We wanted to understand how the mitochondrial proteome has adapted to the loss of this protein. The differentially expressed proteins are identified using the LC/LC-MS technique. The network built using the STRING database provided unbiased identification of network hubs as it builds all the networks entirely on external information. Network enrichment analysis provided several KEGG pathways that were linked to protein machinery, amino acid metabolism, energy production, and fatty acid metabolism.

### 4.1. Proteins Involved in Energy Metabolism

The RMM and WMM tissues had a few proteins following a similar trend in their protein abundance in the icefish. The proteins involved in the ribosomal machinery and cellular hypoxia were found to be commonly increased in the icefish (RMM: ATP5O, ATP5D ATP5B1, ATP5G; WMM: PSMD12, PSMA8). In cluster 1 of the red muscle tissue, apart from involvement in hypoxia, some were also involved in ETC, oxidative phosphorylation, and the TCA cycle (Figure 3A; increased abundance in icefish). ATP synthase subunit O (ATP5O), stress 70 protein (HSPA9), and malate dehydrogenase (MDH1AA) were found to be highly connected to the other nodes, and changes in any highly connected network proteins are likely to be lethal for an organism [45,46,47]. ATP5O is a component of the multi-subunit enzyme ATP synthase (complex V of the electron transport chain), which is located in the stalk that connects the catalytic core (F_1_) to the membrane proton channel (F_O_) [48]. The protein is known to influence the proton conductance by conformational changes [48]. ATP5O has also been found to interact directly with sirtuin 3 (SIRT3), which is significantly involved in energy production and stress responses [49]. ATP5O may contribute to the age-associated decline in association with SIRT3, mitochondria dysfunction, and diseases linked to mitochondrial homeostasis under hypoxia [49,50]. Another component of complex V, ATP synthase subunit gamma (ATPγ), which is also a part of the central stalk, was comparably high in the icefish cohort. ATPγ helps in the binding change mechanism by helping in the rotation of the β subunit. Icefish have been previously reported to show an increased coupling of proton transport and ATP synthase compared with red-blooded notothenioids [51] and this could be correlated to the increase in the specific subunits of complex V that are directly involved in proton translocation. Previously, we have shown the ATP synthase subunit 6 of complex V to be sequentially and structurally different in the icefish *C. gunnari* when compared to its red-blooded related species [52].

One of the proteins densely connected in RMM (a hub node) in cluster 1 also involved in the TCA was the dihydrolipoyllysine-residue acetyltransferase component (DLAT). The increase in the abundance of this protein subunit, a component of the pyruvate dehydrogenase complex (PDH), involved in the breakdown of pyruvate to acetyl-CoA that requires converting NAD^+^ to NADH, in icefish could be a response to moderate pyruvate levels. This could be to prevent pyruvate being converted into acetyl-CoA for TCA and instead be used to meet muscle energetic demands via anaerobic respiration [53,54]. The skeletal muscle is known to have metabolic flexibility in meeting the energy demands of the tissue and responding and adapting to environmental changes [53,55]. The hearts and skeletal muscle of icefish have been suggested to have a dual oxidative-anaerobic metabolism to maintain ATP levels [56,57,58,59]. An increase in the metabolites of fatty acid metabolism in icefish have also been suggested previously [59]. This could further suggest that the increase in DLAT is to maintain pyruvate levels, which could be converted to oxaloacetate for fatty acid cholesterol biosynthesis via the glycolytic pathway [53]. PDH has been seen to be involved in metabolic rate depression in vertebrates, which is a common element of anaerobiosis [60]. The regeneration of NAD^+^ either happens aerobically via OxPhos or anaerobically by fermentation wherein lactate dehydrogenase (LDH) converts pyruvate to lactate. The LDH enzyme has previously been reported to be highly increased in the icefish myocardium, which indicates involvement of anaerobic energetics in the icefish heart muscle [57] LDH is regulated by the relative concentrations of its substrates, and an increase in pyruvate could inhibit the enzyme [61,62,63].

### 4.2. Muscle-Contraction Proteins

A common decrease in the expression in icefish of the proteins associated with biological processes of striated muscle contraction (RMM: MYBPC1, MYBPHA, NEB, TNA3, TTN3A, TNNT3A; WMM: MYBPC1, MYHB, MYBPC2A, MYBPC2B, MYBPC3, TTNA), creatine metabolism (RMM: CKMT2A, CKMB; WMM: CKMT2A), and amino acid/protein metabolism (RMM: GCDH, OGDHA, BCKDHA, PSME2, CKMT2A; WMM: CCT8, VCP, TFA, RPS11, RPS3A) was observed in both RMM and WMM. Creatine kinase (CK), which has previously been shown by Western blot to be absent in the hearts of icefish, did not appear in our protein lists from skeletal muscle mitochondria [26]. The mitochondria are enlarged in the oxidative muscle of icefish, which decreases the distance for the diffusion of oxygen and for ATP between mitochondria and myofibrils, which might reduce the demand for CK [29]. Mitochondrial CK helps maintain flux through the respiratory chain by maintaining low ATP levels [64]. This is coherent with our observation of an increase in the abundance of proteins of complex V. Notothenioids lacking mtCK may compensate by increasing levels of ADP/ATP nucleotide translocases and voltage dependent anion-selective channel proteins (VDACs), as observed in our proteomics study as well, where VDAC3 is selectively seen higher. The lower levels or absence of this enzyme has previously also been reported to be an example of a ‘paedomorphic trait’, a juvenile trait that persists into adulthood. This trait observed as a result of delayed development has been seen as a common feature in icefish; whether this is energetically efficient or not is still debatable [26,65].

A common muscle protein that was found to be downregulated both in RMM and WMM tissues is myosin binding protein C isoforms (MYBPC), which encodes myosin binding protein C. MYBPC is a thick filament associated protein that has both structural and regulatory roles in sarcomere assembly [66]. The protein was seen to decrease four- fold in the icefish *C. gunnari* when compared to the red-blooded species, but the amount was comparable in the icefish *C. rastrospinosus*. MYBPC mutations have been shown to increase the energetic cost of contraction in the muscle, and usually are found to act by reducing the protein content [67,68,69]. The downregulation of this protein in icefish is in contrast to a study that has shown significant upregulation of MYBPC in colder temperatures, which is consistent with the observation of the amount of this protein seen in their closely related red-blooded species [70]. Previously, mutations in MYBPC were seen to be involved in increased cardiac oxidative stress in the mouse model [71]. The other sarcomere protein that was seen significantly less in abundance in icefish was myosin heavy chain b (MYBPH); the activity of this protein has been used as a model to explain the mechanism underlying alterations in skeletal muscle contraction [72]. The ATPase reaction of a muscle fibre is determined by its myosin heavy chain composition, which might be altered with ageing, as seen previously in human skeletal muscle [73]. An impaired sarcomere energetics such as mutations in muscle contractions protein can cause mitochondrial dysfunction due to a Ca^2+^ imbalance or ROS accumulation impairing oxidative phosphorylation capacity [74]. The other proteins that were connected to MYBPH were titin, nebulin, and CK. Titin, nebulin, and CK proteins have previously been seen to be downregulated under long term exposure to hypoxia in the zebrafish model [75]. Chaperonin proteins, T-complex protein 1 encoded by CCT8 are known for their role in the folding of cytoskeleton proteins upon ATP hydrolysis and changes in the protein can cause defects in the functioning of the cytoskeleton and mitosis arrest. A study with *C. elegans* showed CCT8 as a candidate to sustain proteostasis during organismal ageing [76]. The decrease in the level of the protein is observed in human brain ageing and neurodegenerative diseases [77].

### 4.3. Redox

The proteins GCDH- glutaryl-CoA dehydrogenase and Aldh7a1- Alpha aminoadipic semialdehyde dehydrogenase were remarkedly seen lower in the icefish *C. gunnari*, specifically in WMM tissue. GCDH is a mitochondrial enzyme necessary for the metabolism of lysine/tryptophan and hydroxylysine. The absence of this enzyme is known to result in mitochondrial dysfunction. ALDH7A1 is an enzyme that metabolises betaine aldehyde to betaine, which is an important cellular osmolyte and methyl donor that helps in protecting the cell from oxidative stress. The enzyme is seen to be involved in lysine catabolism and helps in maintaining the cellular nitrogen pool [78]. The changes in expression of these proteins might indicate the red-muscle tissue of *N. rossii* and *T. bernacchii* and even *C. rastrospinosus* (that has the expression mb) has enhanced defences against oxidative stress, which is consistent with the previous observation made in the cardiac mitochondrial protein expression data [79]. In WMM, we see that transmembrane protease serine 2-like is very reduced and perhaps missing in icefish. This protein has not been investigated very much to date and has not been reported as absent in these fish previously; the higher levels of it in the red-blooded fish suggest this difference may be of biological relevance.

### 4.4. Haem-Associated Proteins

Consistent with all the studies of icefish, haemoglobin alpha and beta were solely identified in red-blooded species. We also show that hemopexin (Hx) protein is reduced in the WMM of icefish. It was previously shown that Hx transcription occurs at levels comparable to those in the red-blooded notothenioids; however, it seems possible that there is a discrepancy between transcript and protein levels of Hx [80]. Cytochrome c oxidase (CO) is an important haem-containing protein just like haemoglobin and in our study was also found to be much reduced in icefish RMM. As this protein performs a multitude of functions, including cell apoptosis and energy metabolism, it remains to be seen what this reduced level means for the physiology of this tissue [81,82]. The oxygen-carrying capacity of their blood is only 10% compared to that of red-blooded species. Cytochrome c oxidase is at first the terminal electron carrier of the respiratory chain, so lower protein expression may indicate a modification in redox metabolism in icefish. Previously, it has also been shown that maximal capacities of CO and activities of another mitochondrial enzyme, citrate synthase, to be higher in red-blooded fish species in comparison to icefish [83]. Transferrin was another protein that was found to be significantly reduced (five-fold) in the WMM of the icefish *C. gunnari*.

### 4.5. Ribosome Machinery in WMM

Overall, a striking finding in the samples we interrogated is the numbers of 26S, 40S and 60S ribosome proteins that are differentially (mostly) upregulated in icefish. There have been extensive studies on close interaction between mitochondria and the endoplasmic reticulum [84]. FXR2P, a ribosomal binding protein, was also measured at much higher levels in icefish muscle; it would be interesting to probe the connections between ribosome and mitochondrial biology in these organisms. FXR2P protein has not been found to be present in mammalian muscle and therefore its role in icefish muscle remains to be elucidated [85].

With this study, we have established biological pathways and proteins that can be used to understand the unique ‘haemoglobin-free’ biology of icefish. We analysed samples directly taken from the field in the late summer season, ensuring that the pressures of a captive environment are not a factor. We show that muscle mitochondrial proteomes are distinct between fish with different quantities of haemoglobin. We expect our contribution will direct researchers in this field to focus on the identified proteins and pathways that allow these remarkable and unique fish to survive and thrive in Antarctic waters.

## Figures and Tables

**Figure 1 biology-11-01118-f001:**
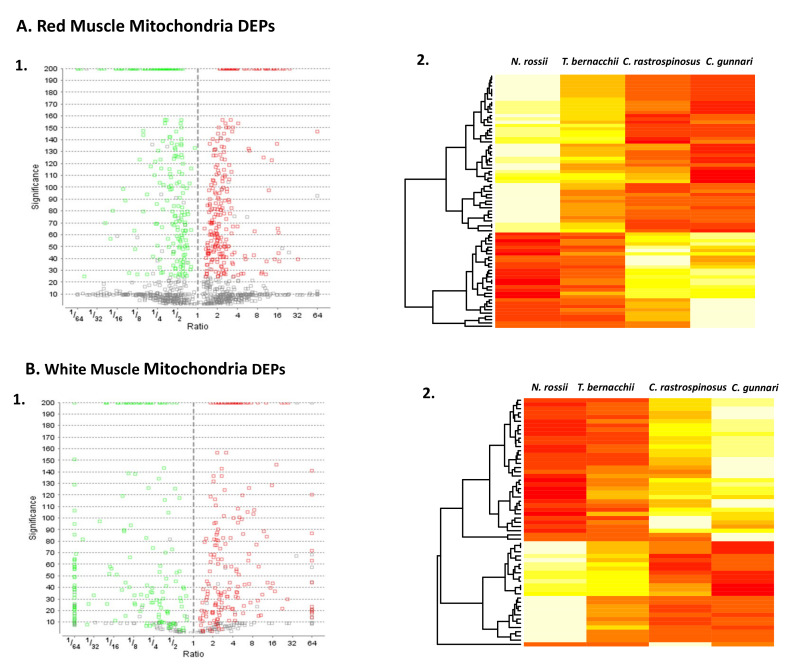
Volcano Plot and Heat Map of DEPs for the two tissue types. Volcano plot for differentially expressed proteins (**A1**,**B1**). The x axis depicts log fold change and y axis depicts FDR; the lower genes are low p values, less significant. Genes that are upregulated are on the right side of the graph and those that down regulate are on the left side of the graph (FDR > 0.01). DEPs in red muscle mitochondria (**A2**) and white muscle mitochondria (**B2**) for the four species (Red is significantly more highly expressed). The heat maps are produced using the relative percent of total ion area that is used for comparing the change in abundance for the same protein from different samples. A clear distinction in the expression of the protein abundance among different species can be seen, red being more abundant and yellow being less (List of proteins Table 1, Table 2, Table 3 and Table 4).

**Figure 2 biology-11-01118-f002:**
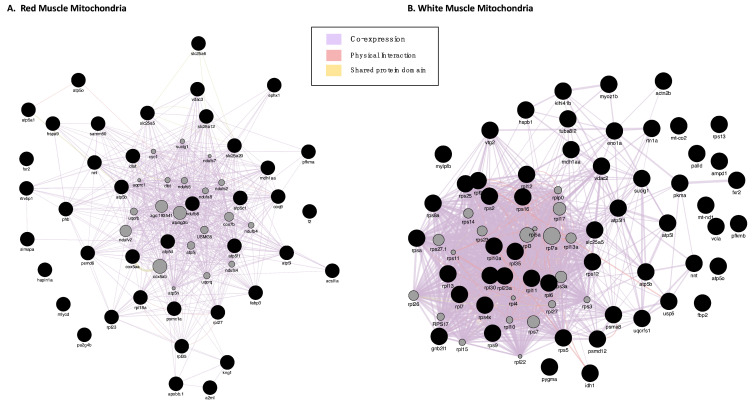
Gene interaction networks for DEPs with increased abudance in icefish. (**A**) RMM Analysis and (**B**) WMM Analysis. A GeneMANIA gene–gene interaction network for protein abundance following the pattern *N. rossii* (+/+), *T. bernacchii* (+/+), *C. rastrospinosus* (−/+) and *C. gunnari* (−/−) in increasing order of their protein abundance, laid out and visualised with Cytoscape, showing interaction strength (edge thickness), interaction type (colour-bottom right), multiple edges between nodes, and protein score (node size). Black nodes indicate query proteins, and grey nodes are neighboring proteins with interactions as co-expressed, physical, or shared protein domains.

**Figure 3 biology-11-01118-f003:**
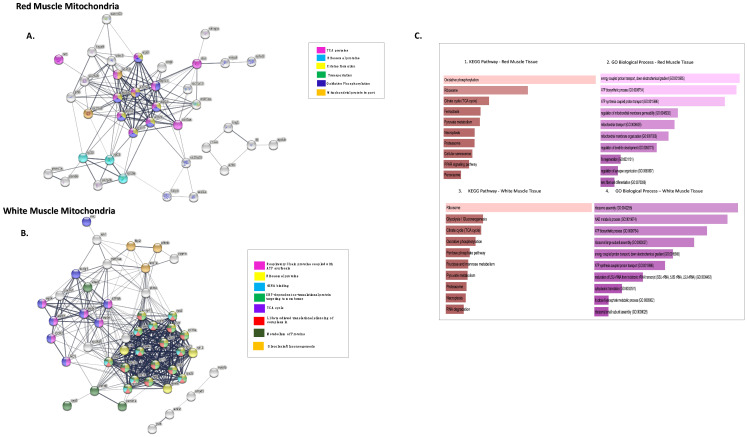
Network analysis of more highly abundant proteins in icefish mitochondria compared with closely related red-blooded species. (**A**) RMM DEPs, number of nodes 42 with 143 edges, the average node degree is 6.8 and (**B**) WMM, number of nodes 56 with 382 edges with an average node degree of 13.6. Network analysed using STRINGdb. The network highlights proteins involved in different pathways curated by STRINGdb from KEGG and Reactome databases. Nodes are coloured according to pathways. The edge shows type of interactions, experimentally determined interactions derived from databases, predicted interactions such as gene neighbourhood, gene co-occurrence, and gene fusions, respectively, co-expression interactions, text-mining interactions, and homology; the thicker the edge, the higher the confidence obtained by the mentioned sources. (**C**) Using FishEnrichr [39,40] analyser KEGG Pathways and GO terms for Biological process showed proteins in different pathways. The length of the bar represents the significance of that specific gene-set or term. Brighter colours are highly significant.

**Figure 4 biology-11-01118-f004:**
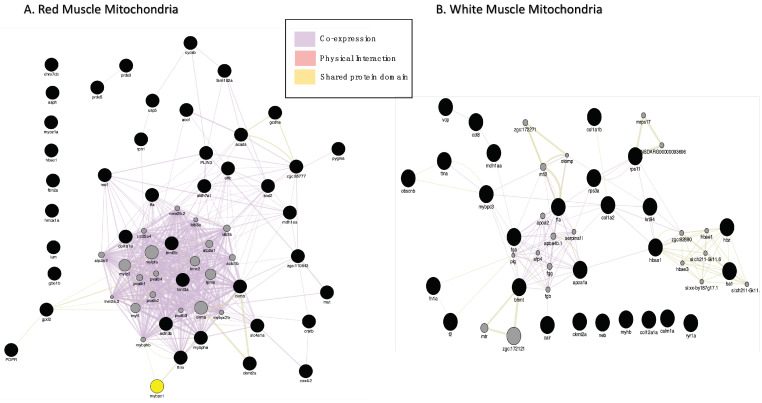
Gene interaction network for proteins with lower abundance in icefish (**A**) RMM and (**B**) WMM. (**A**) GeneMANIA gene–gene interaction network for protein abundance following the pattern *N. rossii* (+/+), *T. bernacchii* (+/+), *C. rastrospinosus* (−/+) and *C. gunnari* (−/−) in decreasing order of their protein abundance, laid out and visualised with Cytoscape, showing interaction strength (edge thickness), interaction type (colour-bottom right), multiple edges between nodes, protein score (node size) Black dots indicate query proteins, and grey dots depict neighbouring proteins.

**Figure 5 biology-11-01118-f005:**
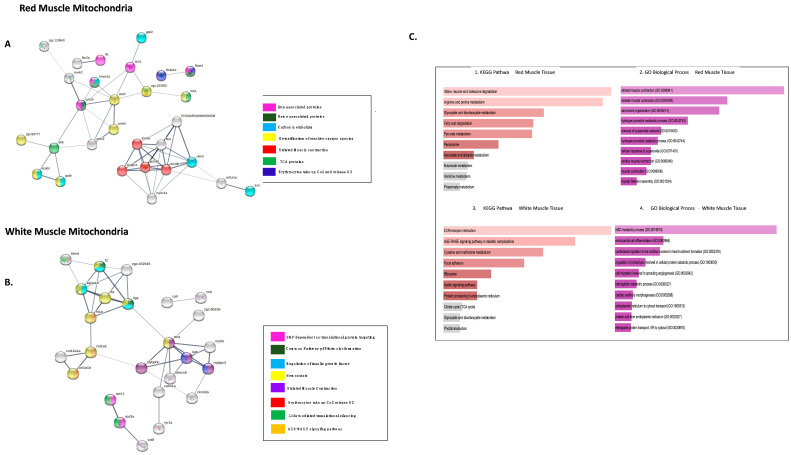
Network analysis of proteins lower expressed in icefish in comparison to closely related red-blooded species. (**A**) RMM DEPs, number of nodes 48 with 53 edges, the average node degree is 2.52 and (**B**) WMM, number of nodes 32 with 45 edges, the average node degree is 2.5. Network analysed using STRINGdb. The network highlights proteins involved in different pathways curated by string from Reactome database. The nodes are coloured according to the pathways determined with Reactome Pathway Database. The edge shows type of interactions, experimentally determined interactions derived from databases, predicted interactions such as gene neighbourhood, gene co-occurrence, and gene fusions, respectively, co-expression interactions, text-mining interactions and homology, thicker the edge higher the confidence obtained the mentioned sources. (C) Using FishEnrichr analyser KEGG Pathway and GO terms for biological process showed proteins involved in different pathways.

**Table 1 biology-11-01118-t001:** Differentially expressed proteins (DEPs) among the four species in RMM—higher abundance in icefish.

S. No.	Protein Abundance—Higher in Icefish (Red Muscle Mitochondria)			Relative Percent of Total Ion Area (Mapped)—Converted from Total Sum of Ion Area to Relative Percent of Total Ion Area				
	Protein Name	Gene Name	Quant Significance H&B Multiple Test Corrected *q*-Value	*N. rossii*	*T. bernacchii*	*C. rastrospinosus*	*C. gunnari*	Accession Number
1	26S protease regulatory subunit 4 isoform X1 & X2	psmc1a	1.25 × 10^−6^	4.9	6.6	7	81.4	XP_010780333.1
2	Apolipoprotein B-100-like partial	apobb	3.76 × 10^−19^	3.2	3.3	27.9	65.6	XP_010781933.1
3	NAD(P) transhydrogenase mitochondrial-like	nnt	2.95 × 10^−19^	5.5	5.8	29.6	59.2	XP_010786020.1
4	Hyaluronan and proteoglycan link protein 1	hapln1	7.26 × 10^−20^	9.1	5.2	30	55.7	XP_010767902.1
5	Sarcolemmal membrane-associated protein-like isoform X1 & X2	slmapa	9.73 × 10^−3^	15.2	15.7	24	45.1	XP_010782086.1
6	ADP/ATP translocase 2-like	slc25a5	1.03 × 10^−18^	13	6.2	36.5	44.3	XP_010765274.1
7	Dihydrolipoyllysine-residue acetyltransferase component of pyruvate dehydrogenase complex	dlat	9.09 × 10^−12^	15.9	16.3	27.3	40.5	XP_010773292.1
8	Alpha-2-macroglobulin-like partial	a2ml	1.23 × 10^−4^	14.7	22.1	23.1	40.1	XP_010771939.1
9	Inter-alpha-trypsin inhibitor heavy chain H4-like	itih3a.2	1.37 × 10^−11^	12.1	12.7	35.9	39.3	XP_010793736.1
10	Stress-70 protein mitochondrial-like	hspa9	1.58 × 10^−5^	20	18.3	33.7	28.1	XP_010766277.1
11	Epoxide hydrolase 1	ephx1l	3.79 × 10^−20^	10.2	21.6	31.1	37.1	XP_010790338.1
12	Sorting and assembly machinery component 50 homolog	samm50	1.65 × 10^−19^	17.4	8.9	37.3	36.4	XP_010773400.1
13	Calcium-binding mitochondrial carrier protein Aralar1	slc25a12	1.21 × 10^−19^	23	7.8	32.8	36.4	XP_010768357.1
14	Long chain fatty acyl CoA synthetase	acsl1a	5.88 × 10^−7^	2.6	19.1	42.1	36.1	AAK07470.1
15	Malonyl-CoA decarboxylase mitochondrial	mlycd	3.73 × 10^−20^	10.9	18.7	34.7	35.6	XP_010792494.1
16	NADH dehydrogenase [ubiquinone] 1 beta subcomplex subunit 8 mitochondrial	ndufb8	6.625 × 10^−7^	13.1	19.5	31.9	35.6	XP_010779169.1
17	60S ribosomal protein L27a isoform X1 & X2	rpl27a	5.17 × 10^−20^	10.7	14.8	38.9	35.6	XP_010790259.1
18	Fatty acid binding protein H8-isoform	fabp3	2.43 × 10^−19^	2.4	25.6	37.2	34.8	AAC60356.1
19	ATP-dependent 6-phosphofructokinase muscle type-like	pfkm	2.02 × 10^−10^	22.3	12.5	31	34.2	XP_010794434.1
20	Carnitine/acylcarnitine carrier protein	slc25a20	1.2 × 10^−19^	20	9.9	36.2	33.9	XP_010773584.1
21	ATP synthase subunit O	atp5o	4.06 × 10^−8^	24.6	13.4	29.1	32.9	XP_010772138.1
22	Proliferation-associated protein 2G4-like	pa2g4b	4.31 × 10^−3^	19	21.8	27.3	32	XP_010782747.1
23	ATP synthase subunit g mitochondrial	atp5l	2.26 × 10^−7^	23.1	10.6	34.4	31.9	XP_010779232.1
24	Voltage-dependent anion-selective channel protein 3	vdac3	4.6 × 10^−19^	13.2	12.1	43.2	31.6	XP_010782516.1
25	ATP synthase subunit alpha mitochondrial	atp5fa1	7.02 × 10^−9^	22.2	14.9	31.7	31.2	XP_010779868.1
26	ATP synthase subunit beta mitochondrial	zgc:163069	1.07 × 10^−8^	24.7	13.4	31.1	30.8	XP_010765728.1
27	ATP synthase subunit gamma mitochondrial isoform X1	atp5g	2.53 × 10^−14^	18.3	10	41.5	30.1	XP_010778067.1
28	60S ribosomal protein L35	rpl35	2.93 × 10^−15^	14.9	15.2	40.2	29.6	XP_010790499.1
29	ATP synthase F(0) complex subunit B1 mitochondrial	atp5pb	1.54 × 10^−15^	23	11.6	36	29.5	XP_010786327.1
30	ATP synthase subunit delta mitochondrial	atp5d	6.63 × 10^−3^	18.8	23.4	28.4	29.4	XP_010775450.1
31	Prothrombin partial	f2	7.41 × 10^−20^	14.3	23.2	33.2	29.3	XP_010786167.1
32	ADP/ATP translocase 3	slc25a6	5.06 × 10^−12^	23.5	13.6	33.9	29	XP_010784438.1
33	60S ribosomal protein L18a-like	rpl18a	3.97 × 10^−8^	18.2	18.8	34.1	28.9	XP_010774792.1
34	ATP synthase subunit d mitochondrial	atp5pd	8.44 × 10^−20^	25.2	8.6	37.5	28.7	XP_010766730.1
35	Malate dehydrogenase cytoplasmic-like partial	mdh1aa	7.88 × 10^−16^	13.2	16.9	42.1	27.8	XP_010766317.1
36	Cytochrome c oxidase subunit 5A mitochondrial isoform X2	cox5a	5.67 × 10^−20^	16.5	9.2	46.4	27.8	XP_010766309.1
37	60S ribosomal protein L23	rpl23	1.13 × 10^−12^	15.9	20.4	37	26.7	XP_010783746.1
38	Prohibitin	phb	2.72 × 10^−6^	18.1	18.4	39.1	24.5	XP_010773724.1
39	Ubiquinone biosynthesis protein COQ9	coq9	8.44 × 10^−11^	17.2	17.6	42.1	23	XP_010793356.1
40	26S proteasome non-ATPase regulatory subunit 6	psmd6	3.76 × 10^−2^	8.5	15	28.3	48.1	XP_010773228.1
41	Fragile X mental retardation syndrome-related protein 2	fxr2	7.06 × 10^−2^	9.6	10.5	16.4	63.5	XP_010770797.1
42	Reticulon-4-interacting protein 1 homolog mitochondrial-like	rtn4ip1	1.04 × 10^−1^	19.8	20.5	28.7	31	XP_010790805.1
43	Kininogen-1	kng1	1.28 × 10^−1^	13.7	16.6	25.8	43.8	XP_010787469.1

**Table 2 biology-11-01118-t002:** DEPs between WMM with increased abundance in icefish specifically seen upregulated in icefish *Champsocephalus gunnari*.

	Protein Abundance—Higher in Icefish (White Muscle)			Relative Percent of Total Ion Area (Mapped)—Converted from Total Sum of Ion Area to Relative Percent of Total Ion Area				
S. No.	Protein Name	Gene Name	Quant Significance H&B Multiple Test Corrected *q*-Value	*N. rossii*	*T. bernacchii*	*C. rastrospinosus*	*C. gunnari*	Accession Number
1	Myosin regulatory light chain 2 skeletal muscle isoform-like	mylpfb	3.42 × 10^−19^	1	2.2	2.3	94.4	XP_010770965.1
2	Myozenin-1 isoform X1 & X2	myoz1b	1.54 × 10^−7^	0	7.8	13.3	78.9	XP_010791910.1
3	Heat shock protein beta 1	hspb1	9.48 × 10^−5^	9.9	4.1	11.5	74.6	XP_010788098.1
4	60S ribosomal protein L35	rpl35	4.58 × 10^−20^	3.4	7.9	32.7	56	XP_010790499.1
5	40S ribosomal protein S16 isoform X1	rps16	9.29 × 10^−3^	12.3	16.9	17.9	52.8	XP_010773777.1
6	60S ribosomal protein L7	rpl7	1.08 × 10^−6^	6.9	7.8	33.1	52.3	XP_010770361.1
7	Fragile X mental retardation syndrome-related protein 2	fxr2	6.55 × 10^−4^	4.6	11.6	32.2	51.6	XP_010770797.1
8	40S ribosomal protein S12	rps12	1.13 × 10^−4^	8.4	13.8	27	50.8	XP_010783785.1
9	60S ribosomal protein L30	rpl30	8.33 × 10^−5^	3.1	22.6	24	50.3	XP_010765856.1
10	60S ribosomal protein L12 isoform X2	rpl12	5.13 × 10^−20^	9.9	11.7	28.5	49.8	XP_010779104.1
11	Reticulon	rtn1a	5.51 × 10^−20^	14.9	13	22.7	49.4	XP_010790870.1
12	Palladin-like	palld	1.07 × 10^−14^	2.7	15.9	32.1	49.2	XP_010785200.1
13	60S ribosomal protein L9	rpl9	4.70 × 10^−20^	4.2	3.4	43.7	48.7	XP_010776310.1
14	AMP deaminase 1 isoform X1 & X2	ampd1	1.65 × 10^−19^	18.6	10.9	22.8	48.4	XP_010793467.1
15	ADP/ATP translocase 3	slc25a5	3.22 × 10^−10^	7.8	10.6	33.7	47.8	XP_010784438.1
16	40S ribosomal protein S13	rps13	9.87 × 10^−4^	13.1	10.6	29.1	47.3	XP_010794693.1
17	40S ribosomal protein S8-like partial	rps8	2.91 × 10^−10^	9.1	11.8	33	46	XP_010787537.1
18	Alpha-actinin-2	actn2	1.42 × 10^−8^	12.1	18.2	23.6	46	XP_010791686.1
19	Kelch-like protein 41b	klhl41b	5.57 × 10^−20^	4.2	11.6	38.5	45.7	XP_010791686.1
20	60S ribosomal protein L6	rpl6	4.536 × 10^−20^	7	11.3	36.3	45.4	XP_010774286.1
21	40S ribosomal protein S5	rps5	7.75 × 10^−4^	17	12.2	25.5	45.3	XP_010782543.1
22	Cytochrome c oxidase subunit II	mt-co2	5.91 × 10^−13^	2.6	26.8	25.9	44.7	XP_010783741.1
23	40S ribosomal protein S25	rps25	6.64 × 10^−3^	13	14.5	28.4	44.2	YP_004581500.1
24	60S ribosomal protein L11	rpl11	4.98 × 10^−20^	15.6	11.7	29.1	43.7	XP_010776714.1
25	Voltage-dependent anion-selective channel protein 2	vdac2	8.76 × 10^−7^	15.6	10.6	30.4	43.5	XP_010779161.1
26	40S ribosomal protein S4	rps4x	6.41 × 10^−20^	7.5	16	33.6	43	XP_010767141.1
27	40S ribosomal protein S2	rps2	5.4 × 10^−20^	6.8	11.4	39	42.8	XP_010792965.1
28	60S ribosomal protein L10a	rp10a	2.26 × 10^−14^	14.8	15.1	29.7	40.4	XP_010783756.1
29	40S ribosomal protein S9	rps9	4.40 × 10^−8^	12.5	17.9	29.3	40.3	XP_010791484.1
30	ATP synthase subunit g	atp5l	4.66 × 10^−20^	18.7	11.6	32.7	37	XP_010786813.1
31	Vinculin	vcla	1.15 × 10^−7^	18.7	11.6	32.7	37	XP_010794136.1
32	60S ribosomal protein L13	rpl13	5.51 × 10^−11^	8.2	16	39.4	36.5	XP_010787927.1
33	Fructose-1 6-bisphosphatase isozyme 2-like	fbp2	5.63 × 10^−20^	9.8	26	28.1	36.1	XP_010789836.1
34	40S ribosomal protein SA isoform X2	rpsa	6.10 × 10^−20^	10	17.7	37.1	35.3	XP_010781656.1
35	Succinyl-CoA ligase	suclg1	6.66 × 10^−11^	5.5	10.2	49.4	35	XP_010768032.1
36	Tubulin alpha chain-like isoform X1 & X2	tuba8l2	4.49 × 10^−4^	11.3	23.9	29.9	34.9	XP_010778226.1
37	60S ribosomal protein L23a	rpl23a	5.29 × 10^−8^	3.5	9.8	56.9	34.8	XP_010766070.1
38	NADH-ubiquinone oxidoreductase	mt-nd1	8.21 × 10^−7^	3.5	9.8	51.9	34.7	XP_010791811.1
39	Alpha-enolase-like	eno1a	4.27 × 10^−4^	13	22.3	30.3	34.5	XP_010777506.1
40	Isocitrate dehydrogenase	idh1	1.22 × 10^−19^	20.6	11	35.7	32.6	XP_010765339.1
41	26S proteasome non-ATPase regulatory subunit 12	psmd12	1.19 × 10^−3^	10	11.8	47.6	30.6	XP_010791048.1
42	ATP synthase F(0) complex subunit B1	atp5pb	2.06 × 10^−10^	19	6.4	44.2	30.2	XP_010777584.1
43	Peptidyl-prolyl cis-trans isomerase-like	pplb	1.02 × 10^−3^	8	27.2	35.3	29.5	XP_010786327.1
44	Malate dehydrogenase	mdh1aa	1.71 × 10^−19^	15.1	9	46.6	29.4	XP_010790691.1
45	ATP synthase subunit O	atp5o	5.16 × 10^−4^	16.1	9.4	45.9	28.6	XP_010780749.1
46	NAD(P) transhydrogenase	nnt	5.81 × 10^−8^	15.9	16.7	39	28.3	XP_010772138.1
47	Glycogen phosphorylase muscle form-like	pygma	8.47 × 10^−6^	17.7	25.3	30	26.9	XP_010776087.1
48	Ubiquitin carboxyl-terminal hydrolase 5 isoform X1	usp5	4.8 × 10^−4^	9.3	19.1	44.8	26.7	XP_010788355.1
49	Cytochrome b-c1 complex subunit 2	uqcrfs1	4.32 × 10^−4^	11.7	10.4	52.9	25	XP_010769500.1
50	ATP synthase subunit gamma	atp5g	1.41 × 10^−12^	13.4	12.3	49.6	24.7	XP_010784571.1
51	ATP synthase subunit beta	zgc:163069	9.98 × 10^−16^	17.5	12.4	46.2	23.8	XP_010778067.1
52	Pyruvate kinase PKM	pkma	5.13 × 10^−19^	11	10.3	56	22.7	XP_010765728.1
53	Vitellogenin-1-like	vtg2	1.38 × 10^−2^	0	2	93.6	4.4	XP_010766216.1
54	Proteasome subunit alpha type-7-like	psma8	1.96 × 10^−2^	9.6	11.7	55.1	23.6	XP_010779640.1
55	Guanine nucleotide-binding protein subunit beta-2-like 1	gnb2l1	1.57 × 10^−2^	13.2	21	29.6	36.2	XP_010783619.1
56	ATP-dependent 6-phosphofructokinase muscle type-like	pfkmb	1.84 × 10^−2^	16.7	17.3	26.1	39.9	XP_010780163.1
57	Inter-alpha-trypsin inhibitor heavy chain H3-like	zgc:110377	1.92 × 10^−2^	0	14.4	46.5	39.1	XP_010782695.1

**Table 3 biology-11-01118-t003:** DEPs in RMM with decreased abundance in icefish specifically seen downregulated in icefish *Champsocephalus gunnari*.

	Protein Abundance—Lower in Icefish (Red Muscle)			Relative Percent of Total Ion Area (Mapped)—Converted from Total Sum of Ion Area to Relative Percent of Total Ion Area				
S. No.	Protein Name	Gene Name	Quant Significance H&B Multiple Test Corrected *q*-Value	*N. rossii*	*T. bernacchii*	*C. rastrospinosus*	*C. gunnari*	Accession Number
1	Hemoglobin subunit alpha-1	hbae1	1.88 × 10^−19^	60	38.7	0.8	0.5	NP_001290227.1
2	Calsequestrin-1	casq1	6.08 × 10^−20^	33.3	44.4	18.6	3.7	XP_010782377.1
3	Perilipin-3	plin3	1.53 × 10^−19^	48.8	35.6	11.1	4.5	XP_010778108.1
4	creatine kinase S-type mitochondrial-like	ckmt2a	4.35 × 10^−19^	71.4	22.4	1.4	4.8	XP_010772488.1
5	Transferrin	tfa	8.36 × 10^−20^	34	54.2	17.3	5.5	CAL92189.1
6	Dehydrogenase/reductase SDR family member 7C	dhrs7cb	3.01 × 10^−9^	68.9	18.2	7.1	5.7	XP_010784042.1
7	Myosin-binding protein H-like	mybpha	9.09 × 10-^20^	66.6	20.7	5.9	6.8	XP_010764981.1
8	Band 3 anion transport protein	slc4a1	1.91 × 10^−8^	39	52	2.1	6.9	XP_010785995.1
9	Cytochrome c oxidase subunit 4 isoform 2 mitochondrial-like	cox4i2	5.79 × 10^−20^	60.1	22.5	9.6	7.7	XP_010770791.1
10	Titin-like	ttna	2.47 × 10^−11^	50.9	21.6	19.4	8.1	XP_010787367.1
11	Glutaryl-CoA dehydrogenase mitochondrial-like	gcdh	4.92 × 10^−20^	34	28.8	28.2	9	XP_010795730.1
12	Fibrillin-1-like isoform X1	fbn2a	5.14 × 10^−20^	67.5	17.5	6	9	XP_010767938.1
13	heme oxygenase 2	hmox1	1.18 × 10^−4^	48.5	20.6	21.3	9.7	XP_010786435.1
14	NADP-dependent malic enzyme	me1	1.29 × 10^−19^	33.9	40.3	16	9.8	XP_010776993.1
15	Pyruvate dehydrogenase phosphatase regulatory subunit	pdpr	8.30 × 10^−9^	24.9	48.6	15.4	11.1	XP_010773093.1
16	Dolichyl-diphosphooligosaccharide--protein glycosyltransferase subunit 1	rpn1	7.17 × 10^−12^	38.5	30.1	20.2	11.2	XP_010777725.1
17	CDGSH iron-sulfur domain-containing protein 1	zgc:110843	4.54 × 10^−20^	41	36	11.6	11.5	XP_010767760.1
18	Glutamate dehydrogenase	gdh	3.06 × 10^−19^	29.9	48.6	11.1	11.9	P82264.1
19	Thioredoxin-dependent peroxide reductase mitochondrial	prdx3	6.46 × 10^−20^	31.2	29.5	27.2	12.1	XP_010779546.1
20	malate dehydrogenase	mdh	1.178 × 10^−12^	32.5	37	18.4	12.1	XP_010765488.1
21	Cytochrome c	cycsb	5.55 × 10^−20^	42.8	28.7	16.1	12.5	XP_010792793.1
22	Troponin T fast skeletal muscle isoforms-like isoform X1 to X3	tnnt3a	4.89 × 10^−20^	37.2	29.6	20.5	12.7	XP_010784864.1
23	Superoxide dismutase [Mn] mitochondrial	sod2	1.01 × 10^−10^	42	33.4	11.5	13.1	XP_010771234.1
24	Carnitine O-acetyltransferase	crat	6.46 × 10^−13^	29.1	38.8	19	13.1	XP_010795330.1
25	PDZ and LIM domain protein 7	ENSDARG00000030638	1.44 × 10^−8^	45.5	21	20.3	13.3	XP_010765699.1
26	Protein FAM162B-like	fam162a	4.13 × 10^−4^	41.5	32	13	13.6	XP_010783349.1
27	Aconitate hydratase mitochondrial	aco1	4.14 × 10^−19^	45.3	26.8	14	14	XP_010781940.1
28	Retinol dehydrogenase 13-like isoform X1 & X2	wu:fd55e03	2.59 × 10^−13^	42.1	26.2	17.6	14.1	XP_010791045.1
29	Myosin-binding protein C slow-type isoform X1 to X17	mybpc	2.39 × 10^−3^	39.8	27.3	18.8	14.1	XP_010774860.1
30	Lumican	lum	2.21 × 10^−7^	47.8	26.9	11.3	14.1	XP_010795529.1
31	Retinol dehydrogenase 13-like isoform X1	rdh13	2.59 × 10^−13^	26.2	42.1	17.6	14.1	XP_010791045.1
32	PDZ and LIM domain protein 7-like isoform X2	ENSDARG00000030638	4.70 × 10^−20^	40.6	26.1	18.9	14.4	XP_010785930.1
33	Glycogen phosphorylase	pygm	7.81 × 10^−20^	30.6	44.1	10.9	14.4	XP_010788472.1
34	Alpha-aminoadipic semialdehyde dehydrogenase	aldh7a1	1.25 × 10^−19^	36.3	28.9	20	14.7	XP_010772035.1
35	Myozenin-1-like	myoz1a	7.07 × 10^−20^	46.5	26.8	12	14.7	XP_010764663.1
36	Collagen alpha-1(I) chain-like	col1a1a	9.40 × 10^−20^	52.6	27.3	5.3	14.8	XP_010768975.1
37	Peroxiredoxin-5 mitochondrial	prdx5	8.32 × 10^−7^	30.5	28.9	25.2	15.4	XP_010783999.1
38	Short-chain specific acyl-CoA dehydrogenase mitochondrial	acads	4.54 × 10^−8^	32.5	26.7	25.1	15.8	XP_010779541.1
39	Troponin alpha-3 chain-like	tnnt3b	1.56 × 10^−19^	36.9	35.9	11	16.1	XP_010771394.1
40	Creatine kinase M-type	ckmb	4.11 × 10^−5^	34.8	25.6	23.3	16.3	XP_010791917.1
41	Glycerol-3-phosphate dehydrogenase mitochondrial	gpd2	1.13 × 10^−10^	42.6	23.5	17	16.9	XP_010791177.1
42	electron transfer flavoprotein subunit beta	etfb	2.38 × 10^−5^	33.9	24.8	24.2	17.1	XP_010791064.1
43	1 4-alpha-glucan-branching enzyme	gbe1b	2.20 × 10^−10^	28.1	44.4	10.4	17.1	XP_010775191.1
44	Ubiquitin carboxyl-terminal hydrolase 5 isoform X1 & X2	usp5	8.40 × 10^−5^	34.9	25.9	21.1	18	XP_010769508.1
45	Methylmalonyl-CoA mutase mitochondrial	mut	2.48 × 10^−4^	33.7	26.7	21.5	18	XP_010784587.1
46	probable acyl-CoA dehydrogenase 6	zgc:85777	1.82 × 10^−3^	31.3	26.9	23.2	18.6	XP_010772948.1
47	Alpha-actinin-3	actn3b	5.52 × 10^−19^	39.5	28.5	13.3	18.7	XP_010784415.1
48	Carbonic anhydrase 1	ca1	7.88 × 10^−20^	49.4	20	10.9	19.6	XP_010765900.1

**Table 4 biology-11-01118-t004:** DEPs between WMM with decreased abundance in the icefish.

	Protein Abundance—Lower in Icefish (White Muscle)			Relative Percent of Total Ion Area (Mapped)—Converted from Total Sum of Ion Area to Relative Percent of Total Ion Area				
S. No.	Protein Name	Gene Name	Quant Significance H&B Multiple Test Corrected *q*-Value	*N. rossii*	*T. bernacchii*	*C. rastrospinosus*	*C. gunnari*	Accession Number
1	Calreticulin	calr	5.72 × 10^−3^	75.4	4.7	19.8	0	XP_010773398.1
2	Creatine kinase S-type mitochondrial-like	ckmt2a	1.19 × 10^−19^	78.2	14.8	6.4	0.7	XP_010772488.1
3	Myosin heavy chain fast skeletal 13	zgc:66156	3.68 × 10^−3^	37.8	34.7	26.6	0.9	XP_010791001.1
4	Prothrombin partial	f2	7.77 × 10^−4^	37.3	30.3	31.2	1.1	XP_010786167.1
5	Fibronectin	fn1a	1.33 × 10^−5^	33.1	30.2	34.9	1.8	XP_010794764.1
6	Transmembrane protease serine 2-like	LOC571565	6.81 × 10^−3^	81.2	16.8	0	2	XP_010778161.1
7	Hemoglobin subunit zeta	hbz	2.98 × 10^−4^	80.5	16.4	0.3	2.5	XP_010778322.1
8	Fibrinogen alpha chain-like	fga	2.67 × 10^−5^	51.4	39.8	5.2	2.8	XP_010771898.1
9	Alpha globin	hbaa1	7.77 × 10^−20^	79.6	16.5	0.4	3.3	AAC25100.1
10	Beta-globin	ba-1	7.45 × 10^−9^	87.5	9.1	0	3.4	AAC60372.1
11	Haemoglobin	hb	7.33 × 10^−9^	87.5	9.1	0	3.4	NP_001290226.1
12	Ryanodine receptor 1-	ryr1a	3.16 × 10^−4^	54.5	19.7	21.1	4.5	XP_010787188.1
13	Keratin type I cytoskeletal 19-like	zgc:153629	8.84 × 10^−20^	54.7	31.8	8.9	4.6	XP_010787448.1
14	Myosin heavy chain fast skeletal muscle-like	myhb	4.58 × 10^−12^	46.7	27.6	19.2	6.5	NP_001290213.1
15	Malate dehydrogenase cytoplasmic	mdh1aa	5.56 × 10^−7^	54.8	24.1	13.3	7.8	XP_010765488.1
16	Apolipoprotein A	apoa1	1.282 × 10^−19^	54.4	20	16.1	9.3	XP_010792180.1
17	Myomesin-1-like	myom1	1.31 × 10^−19^	31.1	29.8	21.8	17.3	XP_010789743.1
18	Betaine-homocysteine S-methyltransferase 1-like	bhmt	7.32 × 10^−20^	53.1	20.8	14.4	11.6	XP_010794476.1
19	Collagen alpha-1(I) chain-like	col1a1b	6.56 × 10^−4^	37.2	34.6	15.8	12.4	XP_010768975.1
20	Hemopexin	zgc:152945	3.14 × 10^−6^	40.7	26.6	19.7	13	XP_010788340.1
21	T-complex protein 1 subunit theta	cct8	3.34 × 10^−6^	60.6	18.2	7.4	13.9	NP_001290219.1
22	Transitional endoplasmic reticulum ATPase-like	vcp	3.10 × 10^−3^	51.4	20.5	12.7	15.4	XP_010770092.1
23	40S ribosomal protein S11	rps11	8.261 × 10^−4^	43.2	32.7	8.7	15.5	XP_010791578.1
24	Titin-like partial	ttna	1.66 × 10^−3^	32.9	28.7	21.1	17.2	XP_010790363.1
25	Calmodulin	calm1a	1.197 × 10^−10^	36.2	30.9	14	18.9	XP_010768524.1
26	40S ribosomal protein S3a	rps3a	9.93 × 10^−12^	35.1	30.1	15.3	19.5	XP_010773841.1
27	Obscurin isoform X2	obscnb	2.1 × 10^−2^	36.8	30.2	14.9	18.1	XP_010790854.1
28	Transferrin	tfa	3.14 × 10^−2^	36.3	33.5	22.7	7.6	CAL92189.1
29	Collagen alpha 1(XII) chain isoform X1, X2, X3, X4 & X5	col12a1a	8.53 × 10^−2^	68.8	18.2	8.5	4.5	XP_010777236.1
30	Nebulin-like isoform X4	neb	2.23 × 10^−1^	54.2	23.8	9.2	12.7	XP_010772593.1
31	Myosin-binding protein C slow-type isoform X1 to X17	mybpc3	1.31 × 10^−1^	45.6	26.9	12.6	14.9	XP_010774870.1
32	Collagen alpha-2(I) chain isoform X1	col1a2	2.91 × 10^−1^	54.6	20.8	9.6	14.9	XP_010772950.1

## Data Availability

Complete mass spectrometry data sets and proteomic identifications are available to download from MassIVE (MSV000089644), [doi:10.25345/C5PZ51Q9F] and ProteomeXchange (PXD034498).

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
