# Peer review of "Quantitative Proteomics and Network Analysis of Differentially Expressed Proteins in Proteomes of Icefish Muscle Mitochondria Compared with Closely Related Red-Blooded Species"

_biology, 2022, doi:10.3390/biology11081118_

Round 1

Reviewer 1 Report

The study involves a mitochondrial protein profiling between the white and red muscle of four closely related fish species using mass spectrometry-based proteomics approach. The study is novel and interesting. The method section is well written. I have only a minor comment that the authors did not validate the differential expression analysis of some of the important proteins by western blot analysis.

Author Response

Thanks for the kind comments about our study.

Having used mass spectrometry to identify and produce relative quantification of the proteins we do not think it is necessary to verify our findings using western blotting which is a far less precise technique.

Reviewer 2 Report

At methodological level the manuscript still has some weaknesses. Mainly concerning network analysis. In addition, it is not acceptable that raw data are not shared. In my opinion the manuscript is not ready for publication

Author Response

Thank you for once again asking us to clarify some points. Reviewer 2 is concerned about some perceived weakness in the analysis but is not particularly specific about what else we should include. There are many papers including in proteomics journals that use cytoscape for networks analyses e.g. Sachez et al., Scientific reports 2020. I can send a long list if that helps.
The complete data have been deposited
Complete mass spectrometry data sets and proteomic identifications are available to download from MassIVE (MSV000089644), [doi:10.25345/C5PZ51Q9F] and ProteomeXchange (PXD034498).

Pre-publication access can be obtained with the details:
ftp://MSV000089644@massive.ucsd.edu
Username = MSV000089644
Password = PHuyJ_-_RfN
Password protection will be removed upon acceptance.

Reviewer 3 Report

The authors now provide full and sufficient description of the Mass Spectrometry procedure including the sample preparation. The reviewer finds the manuscript ready for publication.

Author Response

Thank-you!

This manuscript is a resubmission of an earlier submission. The following is a list of the peer review reports and author responses from that submission.

Round 1

Reviewer 1 Report

The study involves a mitochondrial protein profiling between the white and red muscle of four closely related fish species. The study is novel and interesting. My comments are provided below

Strong Point of the study

  1. The study is more or less complete.
  2. The conclusions drawn are well supported by the result.
  3. The introduction section is well written.

Major weakness of the study

  1. The proteomics data (from Antarctic notothenioid fish ) was compared with another species of fish.
  2. Second, the authors have used Danio for String network analysis.
  3. The authors did not validate the differential expression analysis of important proteins by western blot analysis.
  4. The authors did not discuss the weakness/experimental design of the study.

Other comments

  1. The authors should provide  the cross section of the muscle from these four fish species to ensure that hemoglobin level are different.
  2. The comparison were performed on the mitochondrial enrich fractions. Is there any difference in the mitochondrial density between the four species of fishes?

Reviewer 2 Report

Dear Authors,

I've read with interest your manuscript about the proteomics differences and commonalities of Hb expressing and non-expressing fish species. 

The primary and driving technology for this manuscript is mass spectrometry-based proteomics. It is unfortunate that the manuscript lacks almost completely any relevant details about sample preparation, data acquisition and data analysis. This lack makes it difficult to assess the technical quality of your work, and how well it support the conclusions deduced from it. 

This is extremely important in your case, since you have 1 fish species with what appears to be complete proteome, but 3 species without it. 

The use of PEAKS is understandable under this scenario, but it is unclear (for example) how do you resolve the identification bias of the fish with the sequenced proteome, vs. the species that require homology searches, which should have lower ID rate due to amino acid mismatches. And how to resolve the species that is more distant (with lower ID rate) compared to the species that is closer (which will have  higher ID rate).

Importantly, this will directly affect the quantifications, since quantification is done comparing identical peptides to one another between samples. If you have lower ID rate due to sequence mismatches, you will have artificially reduced the abundance of that protein simply by misidentification.

The problem is also exacerbated since single amino acid mutations (indeed, even small modifications!) can cause vast differences in the ionization efficiency of the peptide the change its intensity in the instrument. thus, two similar peptides that are different by one amino acid can have the same intensity in the instrument and 10 fold difference in reality. 

Unfortunately, non of these vital issues were discussed in the manuscript and the difficulties they present were not addressed. Since the authors didn't submit their data to a public repository such as PRIDE or ProteomeExchange, I couldn't have the chance to try and understand better the data. Data submission is right pre-requisite for proteomics publications for this reason, and reviewer access to this data would help reviewers to put their mind to rest.

Reading the manuscript, I have a strong sense that the proteomics facility which performed the data acquisition and analysis did not influence this publication, as it should - being a proteomics centric paper. this also results in the use of terminologies that as a proteomics investigator of 15 years, I'm unfamiliar with, such as "Relative Percent of Total Ion Area" (of what?!). This feeling is enhanced as the authors did not acknowledge the contribution of the proteomics facility anywhere, despite their (potentially) crucial contribution to the study.

I strongly recommend the authors to involve proteomics specialists, preferably the ones that did the work, in re-writing and reanalysis of the data. As it stands now, it doesn't reach any standard whatsoever in publication of proteomics studies. 

I  strongly encourage the authors to consult the attached guidelines to understand the community requirements for publication of proteomics data:

https://www.mcponline.org/mass-spec-guidelines

Good luck!

Reviewer 3 Report

The manuscript "Quantitative Proteomics and Network Analysis of Differentially Expressed Proteins in proteomes of icefish muscle mitochondria compared with closely related red blooded species" by Katyal et al. aims to shed light on mitochondrial proteome differences among two different icefish species (Champsocephalus gunnari and Chionodraco rastrospinosus) and two related red‐blooded Notothenioids (Notothenia rossii, Trematomus bernacchii), by taking into consideration both red and white muscles. Although the study sounds interesting for the purpose to understand how antarctic icefish mitochondrial proteome has been adapted, it shows relevant missing points concerning the methods used.

In particular:

1) No data on mitochondrial purification/isolation are shown.

2) No details on LC-MS analysis are shown, including protein extraction, amount of proteins analyzed, LC details as well as MS methods applied.

3) Raw data are missing, both in term of mass spectra (they should be deposited in specific database i.e MassIVE) and protein lists with the corresponding statistical parameters (coverage, PSMs, Peal Area, Intensity, P values etc) in each analyzed sample. In this context, please check the sentence in M&M Protein identification: Dissostichus mawsoni (210 sequences; 61,335 residues). As for Dissostichus mawsoni, in both NCBI and UNIPROT we count about 29000 sequences.

4) Network analysis study design is not appropriate:

-The authors build a PPI network model by STRING exploiting the homology with Zebrafish, but no data on sequence homology are shown.

-No information are reported on filters applied in terms of STRING score.

-The authors selected modules by MCODE. No statistical parameters on modules are shown, and it is not clear which kind of interactions (experiment, database, co-expression etc annotated interactions in STRING) they used. Module detection should take in account only physical interactions. In addition, to use co-expression evidences, like in figure2, is poor informative (to state that "In RMM, co‐expression occupied 98.66% of the interactions seen in the network; 263 shared protein domains occupied 1.13%; physical interaction 0.21%"); these percentage differences are due to only the poor coverage of PPIs in Zebrafish.  

-In supplementary figure 2 and 3 the authors show highly connected nodes based in indegree and outdegree data. These are topological parameters associated with Signaling network models (reporting inhibition, activation etc), and not PPI.  They also state that used Centiscape. This data processing, to define hubs, is not described neither cited in M&M. Real and informative hubs should be extracted using different centralities (Betweenness, Centroid, Bridging etc) and validated by Random networks (doi: 10.12688/f1000research.9203.3).

Reviewer 4 Report

The authors investigated the mitochondrial proteome of the antartic icefish and compared it to the one of closely related red-blooded Notothenioids. This careful comparative proteomic study shows that icefish have specific mitochondrial proteome.

  1. The paper lacks a detailed explanation of the Mass Spec instrumentation, and procedure ( including the description of the sample preparation ) without which the sensitivity of the system and experimental reproducibility stays unclear.